



# Beyond precipitation: diversity of drivers of high river flows in European near-natural catchments

Manal Lam'barki[1,*], Wantong Li[1,*], Sungmin O[2], Chunhui Zhan[1], and Rene Orth[1]

[1]Department of Biogeochemical Integration, Max Planck Institute for Biogeochemistry, Jena, 07745, Germany
[2]Department of Climate and Energy System Engineering, Ewha Womans University, Seoul, South Korea

* These authors contributed equally to this work.

*Correspondence to*: Wantong Li (wantong@bgc-jena.mpg.de)

**Abstract.** High streamflow in rivers can lead to flooding, which may have severe impacts on economy, society and ecosystems. Therefore it is imperative to understand their underlying physical mechanisms. Previous research has illustrated the relevance of several hydrological drivers, such as precipitation, snowmelt and soil moisture. However, the relative importance of these drivers compared with each other is unclear. Moreover, the role of vegetation-related drivers is not well studied. In this study, we focus on high river flows and consider a comprehensive set of potential drivers and analyze their relative importance. This is done with streamflow observations from over 250 near-natural catchments located across Europe during 1984-2007, which are matched with driver data from various observation-based sources. Not surprisingly, we find that precipitation is the most relevant driver of high river flows in most catchments. In addition, and more interestingly, we show that next to precipitation a diversity of other drivers is relevant for high flows, including shallow soil moisture, deep soil moisture, snowmelt, evapotranspiration and leaf area index. These non-precipitation drivers tend to be even more relevant for more extreme high flows. The relative importance of most considered drivers is similar across daily, weekly and monthly time scales. The spatial patterns of the relevance of precipitation, snowmelt and soil moisture for supporting high river flows are controlled by vegetation types and terrain characteristics, while climate and basin area are less important. By analyzing a comprehensive selection of drivers of high river flow in a powerful framework which accounts for co-linearities between drivers, this study advances the understanding of flood generation processes and informs respective model development.



## 1. Introduction

Hydrological extremes have significant impacts on society and ecosystems (Kundzewicz and Kaczmarek, 2000; Alfieri et al. , 2020; Orth et al. , 2022; Merz et al. , 2021; Bradford and Heinonen, 2008). For example, droughts and floods have been more devastating than other natural hazards in terms of their socio-economic damage (Barredo, 2007; Naumann et al., 2015; Gao et al., 2019). Knowledge about flood generation mechanisms is key to optimize flood management and protection strategies to mitigate impacts (Merz et al., 2021).

Most major floods are characterized by a synergistic combination of atmospheric circulation patterns delivering large amounts of precipitation, and antecedent basin properties that condition the climate-runoff relationship (Hirschboeck, 1991; Liu, 2019). Therefore, river flooding remains complex to understand as it is not exclusively linked with heavy precipitation but also depends on other factors such as antecedent soil conditions or snowmelt (Berghuijs et al., 2016a; Bertola et al., 2020). For example, soil moisture excess has been shown to be the most relevant hydroclimatic variable to explain flood seasonality in Western Europe (Berghuijs et al., 2019). It has also been shown that wet antecedent soil moisture amplified the floods in the upper Danube in June 2013 (Blöschl et al., 2013).

An analysis of literature in the Web of Science (www.webofscience.com) reveals the focus of recent flood research; flood-related articles often refer to precipitation (19'556 articles during 2002-2021, see Table S1 for detailed search commands), sometimes to vegetation (7'066 articles), and relatively rarely to snow (2'813 articles) and soil moisture (2'804 articles). There are only 11 articles referring to all these drivers simultaneously. And in these 11 articles the focus is mainly on regional and/or modelling studies, and they use some drivers for explanation of the results rather than including them in the actual analysis. This leaves a knowledge gap in the joint understanding of a variety of observation-based controls of high river flows across continental-scale areas. Also, this highlights that it is important to extend the focus towards jointly investigating a multitude of potential drivers of extremes (Brunner et al., 2021b), especially in the context of climate change where increasing precipitation may not necessarily translate to increasing streamflow (Sharma et al., 2018; Brunner et al., 2021a). Moreover, the consideration of several drivers across many catchments allows to analyze the spatial variability in the relevance of individual drivers of high river flows. This way, it is possible to determine which climate, terrain or vegetation characteristics influence these spatial patterns.

Moreover, flood generation processes not only vary across catchments but also vary across different time scales. Previous studies have recognized this by separating different kinds of floods such as flash floods, short rain floods, long rain floods, excess rainfall floods, rain/snowmelt floods, and snowmelt floods (Merz and Blöschl, 2003; Sikorska et al., 2015; Stein et al., 2020; Tarasova et al., 2019). This way, different levels of streamflow may result from similar amounts of precipitation or snowmelt depending on the time scale during which they hit a catchment. This is further modulated by the soil moisture and vegetation conditions during the respective time frame. Many flood-related studies have employed a weekly time scale to infer potential flood drivers (e.g., Berghuijs et al., 2016a; Blöschl et al., 2017; Stein et al., 2020; Tramblay et al., 2021; Wasko et al., 2020), while the relative relevance of flood drivers at different time scales (daily, monthly) remains more unclear. This illustrates the importance of jointly considering different time scales in the analysis of high river flows, in particular because floods as rare extreme events are likely induced by a similarly rare combination of processes or drivers acting across time scales.

The objective of this study is to determine relevant drivers of high river flows in Europe across different time scales in a data-driven way. This is done by jointly analyzing the relationship of high river flows with a multitude of drivers in a comprehensive statistical framework which can account for co-linearities between drivers and for mismatches between the river flow and driver dynamics. Our selection of drivers is based on physical linkages with the land water balance and river flows, and includes vegetation-related variables such as evapotranspiration and leaf area index reflecting interception capacity. As shallow and deep soil moisture might differently affect baseflow and overland flow, we also consider soil moisture separately from different layers. Note that this analysis focuses on high river flows rather than actual floods. While there might be a strong





correspondence between them, streamflow data are more accurate and abundant and hence employed in this study to characterize high flow events. Finally we attribute the determined spatial patterns of the relevance of the main drivers of high flows to vegetation, terrain, catchment and climate characteristics in order to advance the understanding of flood generation processes and to inform hydrological model development.

## 2. Data and Methodology

### 2.1. Data

#### 2.1.1. Streamflow data

We use daily streamflow observations during 1984-2007 obtained from 436 river gauging stations from Stahl et al. 2010 who consolidate data from UNESCO's European Water Archive, regional and national agencies and the EU WATCH (WATer and global CHange) project. These data have been employed and validated in various previous studies, e.g. to build a European flood database (Hall et al., 2015), to empirically evaluate streamflow trends in Europe (Stahl et al., 2010), to analyze water storage sensitivity to streamflow (Berghuijs et al., 2016b), and to estimate continental-scale runoff (Gudmundsson and Seneviratne, 2015). Our study focuses on high flows as determined from high quantiles of daily streamflow. These extreme river flows can potentially coincide with flooding events where water overtops the river channel.

#### 2.1.2. Hydro-meteorological and dynamic vegetation data

We use hydro-meteorological and vegetation-related variables from various sources as potential drivers of high river flows (see Table 1). We focus on drivers with a physical link to streamflow. As these datasets are gridded we are matching the grid cells with the locations of the catchments from which we have streamflow measurements using the method of the nearest neighbor. Then, time series are obtained from the respective grid cell and jointly analyzed with the corresponding streamflow observations. The set of considered drivers includes for example vertically resolved soil moisture, evapotranspiration and leaf area index. The latter is included as a proxy for interception, and its daily estimates are calculated by linear interpolation of monthly values to avoid gaps related to missing daily satellite information due to cloud cover. Further, we take into account precipitation variability by calculating the ratio between the peak daily precipitation and the cumulative precipitation during the considered time scale/window (hereafter referred to as distribution of rainfall).

Daily snowmelt is obtained using the Simple Water Balance Model (Orth and Seneviratne, 2015), using observation-based forcing data Therein, snowmelt is estimated as water equivalent using a degree-day approach; whenever precipitation occurs in combination with a temperature below threshold, snow is formed and stored until temperatures rise above the threshold where the snow is assumed to melt proportionally to the temperature difference to the threshold (Orth and Seneviratne; 2013, 2015). The model is forced with net radiation from the ERA-5 dataset (Hersbach et al., 2020) and precipitation and air temperature from the E-OBS dataset. This is done for the grid cells corresponding to the considered 436 catchments during the study time period. Calibration parameter values are used from Fallah et al. (2020) who calibrated the model against streamflow observations.

The data of potential drivers of high flows are mostly considered from grid cells of 0.25˚ spatial resolution such that they represent an area of approximately 625 km$^2$ which is roughly similar to the mean size of the considered catchments.

#### 2.1.3. Static datasets

To attribute the identified spatial patterns of the relevance of the considered drivers, we consider a range of static data. This includes a characterization of the climate through (i) the long-term mean temperature as inferred from E-OBS data, and (ii) the aridity index as computed from the ratio of average net solar radiation (in MJ.m$^{-2}$.day$^{-1}$) and unit-adjusted precipitation (in mm.day$^{-1}$) from the ERA-5 dataset. We further use the tree cover fraction which is obtained from the VCF5KYR data product and corresponds to the proportion of the ground covered by the vertical projection of tree crowns (Song et al., 2018). We also include catchment attributes such



as basin area and to characterize the terrain we consider mean elevation and slope, which are obtained from a digital elevation model with an original spatial resolution of 250 m (Amatulli et al., 2018) which we aggregated to 0.25° spatial resolution.

**Table 1: Overview of considered drivers of high flow.**

| Variable | Dataset | Type | Unit | Temporal resolution | Spatial resolution | References |
|---|---|---|---|---|---|---|
| Precipitation and precipitation variability (see text) | E-OBS gridded data (v.20) | Interpolated from station observations | mm | | 0.25°×0.25° | Cornes et al. (2018) |
| Evapo-transpiration | Global Land Evaporation Amsterdam Model (v3.5a) | Model-based | mm | daily | | Martens et al. (2016) |
| Snowmelt | Simple Water Balance Model | Model-based | mm | | | Orth and Seneviratne (2015) |
| Soil moisture from multiple depths (layer 1 : 0 - 7cm , layer 2: 7 - 28 cm , layer 3: 28 - 100 cm) | European ReAnalysis 5 (ERA-5) | Reanalysis | $m^3 \, m^{-3}$ | | | Hersbach et al. (2020) |
| Leaf Area Index | GEOV2-AVHRR | Derived from satellite observations | - | monthly interpolated to daily | 0.5° x 0.5° | Verger (CNES-Theia , 2014) |

### 2.2. Methodology

#### 2.2.1. Catchment selection

This study aims to focus on near-natural catchments with no or minor disturbance on river flow due to human intervention. We assess this through the reproducibility of streamflow dynamics by a conceptual model. Using the Simple Water Balance Model (from Orth and Seneviratne; 2013, 2015) and calibration parameters obtained from Fallah et al. (2020), we obtain modeled runoff for each individual catchment. Then we calculate the level of agreement between modeled and observed streamflow for each catchment and disregard catchments where
the agreement is weaker than a threshold score for the Nash-Sutcliffe efficiency of 0.36, which is adopted from O et al, 2020. We assume that in these catchments the streamflow dynamics are affected by non-natural processes such as irrigation or other forms of human management. Further, catchments with more than 10% of missing runoff data are excluded.

As a result, 251 near-natural catchments from 12 countries are selected (Fig. 1). Basin sizes range from 7 to

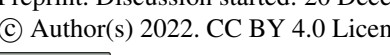



3780 km$^2$, however, with only a small number of catchments (23) with a basin size greater than 1000 km$^2$. These catchments are located across the European continent with fewer samples in the East and the South but nevertheless spanning a considerable climate gradient.

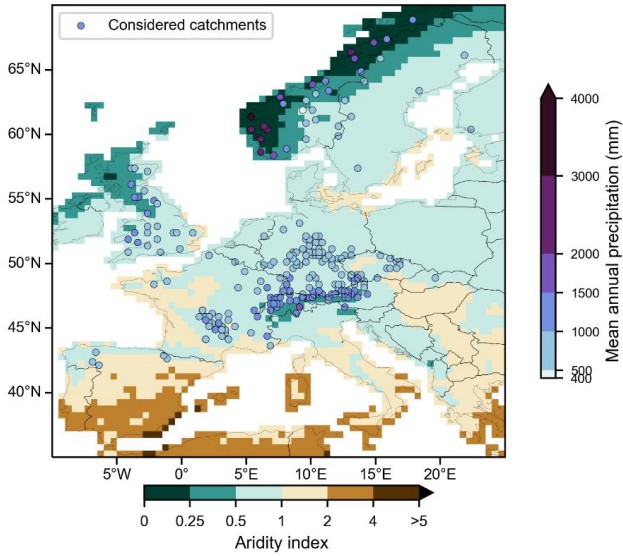

**Figure 1: Locations and climate conditions of the considered near-natural catchments.**

### 2.2.2. Identification of high flow events

To study potential drivers in generating high flows with different magnitudes, we select high flow events from daily streamflow records of each catchment which exceed different thresholds. In particular, we consider the 90th, 95th, 98th, 99th, and 99.5th percentiles of the entire daily streamflow time series of each catchment. For each threshold, we ensure to select high flows which are independent from one another by considering only the daily streamflow peaks which are at least one month apart from each other. The number of selected events for each catchment and percentile threshold and their corresponding magnitudes are shown in Fig. S1.

### 2.2.3. Deriving high flow drivers across time scales

After selecting high flow events, the main drivers of these events are computed for each catchment considering the variables listed in Table 1.

First we remove the mean seasonal cycles from both the streamflow and driver data in order to focus on anomalies. We assume that society and ecosystems in each catchment are adapted to the usual streamflow evolution (i.e. mean seasonal cycle) and most affected by strong deviations from this. The mean seasonal cycles are determined for each variable by averaging values from the same day-of-year across all available years (e.g.., the mean seasonal temperature on the 1$^{st}$ of January corresponds to the average of temperature the 1$^{st}$ of January in each individual year between 1984 and 2007). To remove random variations in the computed mean season cycle, a smoothing function of the calculated seasonal cycle is performed using a centered moving average including 5 previous and subsequent values to calculate the average at each time step.

A novel aspect of our study is the consideration of different time scales in the determination of relevant high flow drivers. For this purpose, we average the driver data anomalies across weekly (7 days) and monthly (30



days) time windows which are positioned before each high flow event. Results for the daily time scale are derived using the concurrent driver data at the day of the high flow.

### 2.2.4. Quantifying the importance of potential drivers of high flows

An overview of our workflow is shown in Fig. 2. For each catchment and each high flow magnitude we consider
all detected daily high flow values together with the corresponding driver anomaly values. This is done separately for each of the considered time scales. We evaluate the relevance of each considered driver using the dredge function from the MuMIn package in the R programming environment (Barton, 2014), which generates numerous multivariate linear regression models considering all possible combinations of considered drivers to predict the considered high flows. These models are then ranked according to their prediction performance using
Akaike's information criterion (AIC) which takes into account the goodness of fit together with the complexity (i.e., number of involved drivers) of each model. Then, we select all models of which the difference between the AIC and the AIC of the best model is less than 4 (as in Denissen et al. 2022). This step ensures that the influence of co-linearities between drivers on our results is minimized which could otherwise lead to inaccurate estimations of their relevance; Models with correlated drivers tend to have less beneficial AIC scores as the
overlapping information content of co-varying drivers reduces the model performance normalized by the considered number of drivers.

Since drivers are selected such as they would be physically linked to runoff, we disregard regression models where the slope between runoff and evapotranspiration or leaf area index is positive, as this indicates confounding effects where e.g. precipitation increases evapotranspiration (and also leaf area index) and runoff at
the same time such that evapotranspiration (and leaf area index) is not an actual driver. If all selected multivariate regression models exhibit estimated positive slopes between runoff and evapotranspiration or leaf area index, we re-compute the multivariate regression analysis for the given catchment without consideration of evapotranspiration or leaf area index as potential drivers of high flows.

As a next step, from the remaining regression models in each catchment we select models with sufficient
predictive power (adjusted $R^2 > 0.3$) to ensure that the contained drivers can actually explain the high flow variability. This also serves as a test of the agreement between the independent streamflow and driver datasets. From the remaining multivariate regression models in each catchment we determine the most relevant driver. This is done by computing the average of the high flow variance explained by each driver across regression models, weighted by the model's AICs. Then, by comparing the fractions of high flow variance explained by
each driver, we determine the most and second-most relevant drivers in each catchment.

We further apply an alternative methodology where we compute Spearman correlations between the selected high flow anomalies and the respective driver anomaly values in each catchment. Also this analysis is done separately for each considered time scale. The most relevant driver is then determined by the highest correlation coefficient. Similarly to the other methodology, positive correlations between evapotranspiration and runoff, and
between leaf area index and runoff are not considered.





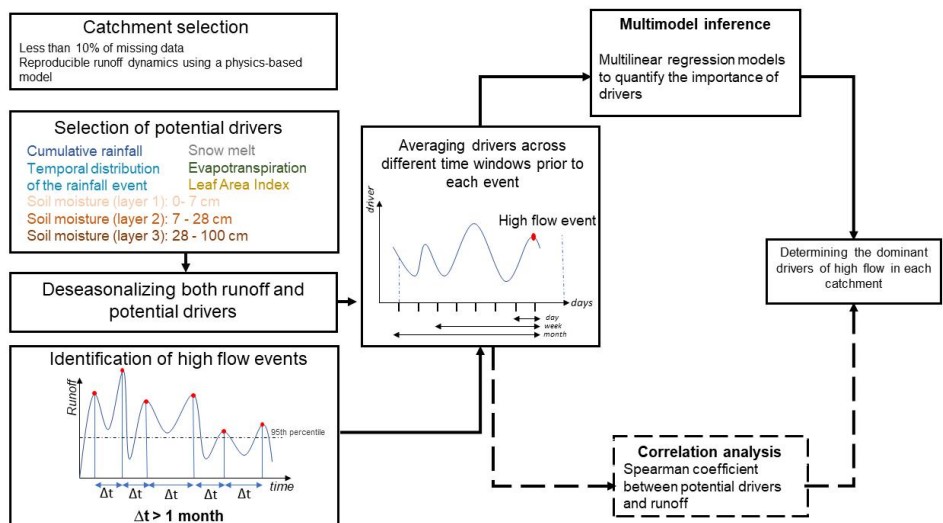

**Figure 2: Illustration of our workflow.** We use runoff data from selected catchments in conjunction with time series of potential drivers physically related to runoff. Independent high flow events are identified from the runoff time series in each catchment, and time series of considered drivers' anomalies are averaged across different time windows 5  before each event. Finally, we investigate relationships between the averaged driver values and the high flow magnitudes using two independent methodologies.

### 2.2.5. Attribution analysis

We study the controls of the spatial patterns of the importance of main high flow drivers, namely precipitation, snow melt and soil moisture. This is done by correlating the driver importances from all catchments with the 10  considered attribution controls including climate, vegetation and terrain parameters. Thereby, we use partial correlations to mitigate the effect of co-linearities between attribution controls and to isolate the individual effect of each control. For the driver importances we apply a normalization to make them comparable across catchments by using a ratio of explained variance of the driver and the average $R^2$ (which is the total explained variance) of all finally considered regression models for each catchment. In the case of soil moisture, we add the 15  explained fractions of high flow variance from all three layers.

20



## 3. Results and discussion

### 3.1. Determining dominant drivers of high flows across time scales and flow magnitudes

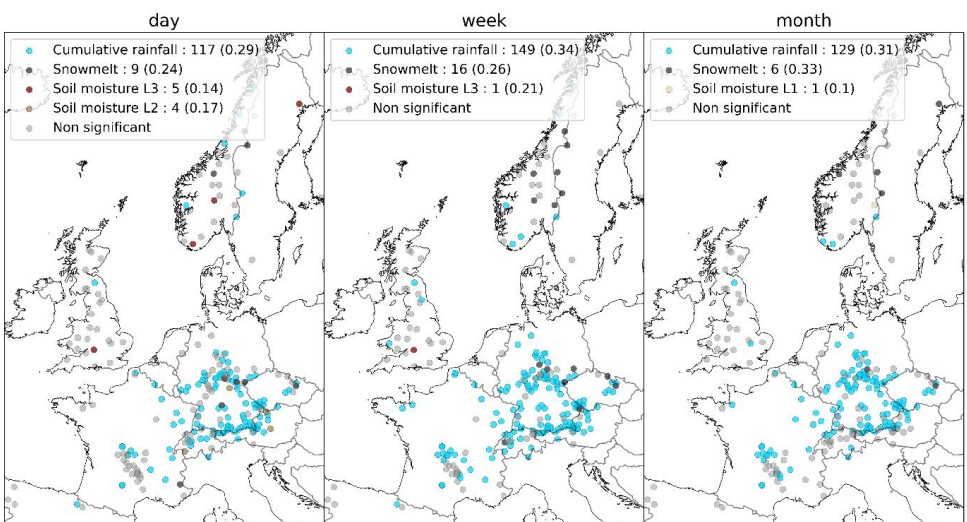

**Figure 3: Illustration of the most influential driver of high flows in each catchment as shown through color-coding. In each catchment all independent high flow events exceeding the 90th percentile are considered, in the three time scales (daily, weekly, monthly) that precede the events. Statistics in the legend indicate the number of catchments where each driver is most influential, and the respective mean fraction of explained high flow variance across these catchments. Non-significant results correspond to the catchments where the average $R^2$ of the models is below 0.3 or if runoff in all the models has a positive relationship with evapotranspiration or leaf area index.**

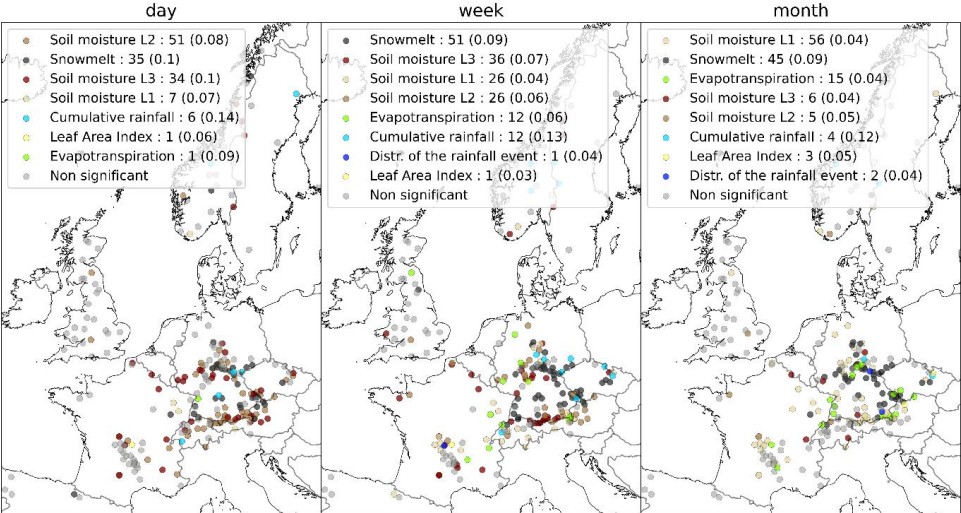

**Figure 4: Illustration of the second most influential drivers of extreme high flows exceeding the 90th percentile. Similar format as in Fig. 3.**





The most relevant drivers of high flows as determined with the multimodel inference approach are shown for each catchment in Fig. 3. Antecedent precipitation is the most relevant driver in the majority of the considered European catchments. This is observed for all considered time scales, and most pronounced at the weekly time scale. This is also demonstrated by the mean explained variance of high flows given in the legend, which varies

between 0.29 and 0.34. Snowmelt is overall the second most relevant driver of high flows, both in terms of the number of catchments where it is the most relevant predictor of high flows, and the relatively large explained fraction of variance of high flows. Snow melt is most relevant in the Alps, the Massif Central in France, and across (other) uplands of Central Europe, and its importance peaks at the weekly time scale. Catchments where no main driver could be determined as no regression model was left after all filtering steps (in particular after

the $R^2 > 0.3$ filtering) are shown in gray.

In addition, we also investigate extreme high flows exceeding the 95th, 98th and 99th percentile (Fig. S2). The results are similar to the findings in Fig. 3 with antecedent precipitation is the most relevant driver of high flows. However, there are more catchments with most relevant drivers of high flow other than precipitation such that the diversity of most relevant drivers is overall enhanced towards more extreme high flows. For example, soil

moisture and the distribution of the precipitation across the considered time scale emerge as most relevant drivers in some catchments.

Another interesting result is that the explained variance of high flows of the dominant drivers is similar across time scales. This indicates that studying drivers at different time scales is relevant to understand high flow dynamics, whereas daily, weekly and monthly time scales are similarly important. Multilayer soil moisture has a

higher explained variance for events of the 99th percentile, suggesting the soil water storage is more relevant for the more extreme high flow generation.

The spatial patterns of most important drivers in Fig. 3 are confirmed with a methodology based on correlations between high flows and drivers (see section 2.2.4) as shown in Fig. S3. While the obtained correlations are highly significant for the results of the 90th percentile high flow threshold, this is less the case for the higher

thresholds as the number of considered high flow events decreases (Fig. S4).

Although the most important high flow driver, antecedent precipitation, is consistent across many catchments, high flow magnitudes, and time windows, the second-most important drivers are generally more diverse, as illustrated in Fig. 4. This diversity is even increasing towards more extreme events (Fig. S5). This indicates the difficulty to understand extreme high flow generation, and highlights the essentials of considering multifaceted

controls of high flow generation. Interestingly, Figure 4 also shows that evapotranspiration and surface soil moisture become more relevant towards longer time scales while deep soil moisture gets less relevant. In the case of evapotranspiration this is probably related to the fact that this becomes larger when aggregated across longer time periods. The shift in the relevance of shallow versus deeper soil moisture with increasing time scales could be explained with stronger precipitation events at short time scales which might saturate the surface soil

such that the soil moisture in deeper layers becomes more relevant to buffer or enhance the resulting streamflow. By contrast, at longer time scales the precipitation is typically distributed to several events which might not be sufficient to saturate the surface soil such that this layer then determines more which fraction of the precipitation contributes to streamflow.



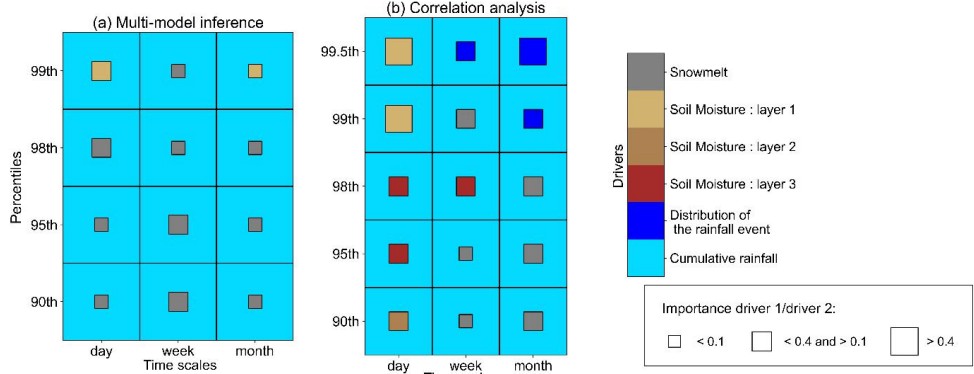

**Figure 5: Summary of our results. Box colors indicate the most influential drivers across time scales and high flow magnitudes, as determined from the number of catchments where a particular driver is found to be most influential. Second most relevant drivers are shown within each box through color-coding. Results shown for (a) multi-model regression inference methodology and (b) an independent correlation analysis. Note that results for the 99.5th percentile could not be computed for the multi-model inference approach due to the low number of such extreme high flows.**

Next, we summarize the results of the most and second-most relevant drivers of high flows across time scales and high flow magnitudes in Fig. 5. Results are shown for both considered methodologies of determining the most relevant high flow drivers. Both methods demonstrate that antecedent precipitation is the dominant driver of high flows across all considered time scales and flow magnitudes. Snowmelt is overall the second most important driver. Note that as soil moisture layers are considered separately the overall relevance of soil moisture might be underestimated in Fig. 5. However, the results in Fig. 3 and S3 in terms of the number of catchments where snow melt or any soil moisture layer is the most relevant high flow driver do not indicate a strong bias in the results of Figure 5.

The correlation analysis generally supports the multi-model inference results, even though soil moisture is found as a second-most important driver for many high flow magnitudes at the daily time scale instead of snowmelt. The correlation analysis allows to compute results for very extreme high flows exceeding the 99.5th percentile while the multimodel inference method does not detect any suitable regression models which can be fitted for such few remaining high flow events in most catchments. For such extreme high flows the distribution of rainfall and soil moisture become more relevant in a greater number of catchments, even though still less catchments where precipitation is found to be most relevant. These results are, however, not statistically significant due to the low number of considered extreme high flows. Our results therefore confirm previous studies which have demonstrated that river floods are usually generated by the interactions between event precipitation, antecedent soil wetness, and snowmelt (Merz and Blöschl, 2003; Tarasova et al., 2019). The current study additionally shows that a multitude of drivers other than precipitation become increasingly relevant towards more extreme high flows.



### 3.2. Attribution analysis



**Figure 6: Attributing the spatial patterns of the relevance of considered drivers of high flows to climate, vegetation and terrain characteristics for high flow events exceeding the 90th percentile. Vertical axis corresponds to the partial correlation between driver relevance and each attribution variable. Results are shown for the considered different time scales. Stars on top of the bars indicate statistically significant partial correlations (\*\* : p-value <0.05 , \*\*\* : p-value < 0.005).**

We furthermore study the controls of the spatial variations of the relevance of main high flow drivers.

Figure S6 shows the relevance of precipitation in each catchment, soil moisture and snowmelt.

As described in section 2.2.5 we consider climate, basin and terrain attributes as potential controls in this context. Figure 6 provides the results and shows that tree over fraction is overall most important in explaining spatial patterns of the relevance of precipitation, snowmelt and soil moisture for high flows. It remains also an important control when considering only the tree cover fraction, the elevation and the slope (Fig. S7).

In more tree-covered regions the relevance of precipitation for causing high flows tends to increase while that of

snowmelt and soil moisture tends to decrease. This might be related to litterfall which impedes the infiltration of water into soils and hence increases the fraction of precipitation contributing directly to streamflow, while the contributions of soil moisture and snowmelt are decreased. We find an increased relevance of precipitation in warmer catchments which is probably related to the higher rain-to-snow ratio. By contrast the results for slope and elevation are hard to interpret and further research, potentially with more diverse catchments offering more

variability in terms of slope and elevation allowing to derive more informative and significant results.

In general, the results for precipitation are opposite to those of snowmelt and soil moisture, indicating that whenever a considered control favors precipitation this comes at the expense of the relevance of snowmelt and soil moisture, and vice versa.



### 3.3. Limitations

While we test the robustness of our results with two independent methods, the main findings have to be seen in the light of some limitations related to our data and methodology. First, there is a spatial mismatch between the catchment area and the grid cells from which the driver data is derived. While there is an overlap between the different regions, the time series do not exactly represent the same areas. However, in most catchments the employed driver data corresponds sufficiently well with the observed high flow dynamics as tested with the $R^2$ threshold in the multimodel inference approach such that there seems to be a sufficient level of agreement between the considered data streams. This could in principle be different for smaller versus larger catchments but the attribution analysis indicated that the results do not vary according to catchment size.

Second, it is possible that trends in the considered data streams could influence our results and induce shifts in the relevance of high flow drivers over time. However, the visual inspection of the streamflow time series in many catchments does not indicate trends in our target variable such that this should not affect our results.

Third, even though we are considering a comprehensive set of potential drivers of high river flows there might be more influential drivers representing alternative processes which are not captured by our analysis. This applies for example to groundwater which we could not include here due to a lack of sufficient data.

Finally, the attribution analysis is somewhat limited by the fact that only European catchments are considered here such that the spatial variability of climate, vegetation and terrain characteristics is rather low. Future research focusing on larger sets of catchments with more diversity in these aspects could provide more significant insights into the spatial variations of the relevance of main flood drivers.

### 4. Conclusion

This study provides a quantitative mapping of the importance of drivers of high river flow in near-natural European catchments. We consider a comprehensive set of drivers, and use a powerful statistical approach based on multiple multivariate regressions to determine their relative importance across time scales and high flow magnitudes. In agreement with previous knowledge and literature, we find that antecedent precipitation anomalies are the most important driver of high flows in most catchments. In some other catchments snowmelt and soil moisture are found to be the most relevant drivers. Moving beyond the state of the art we find a remarkable diversity of second-most important drivers across Europe. This includes vegetation-related drivers such as evapotranspiration. Overall, observed daily high flow dynamics can be explained similarly well using drivers from the daily, weekly and monthly time scales. This indicates that mechanisms acting at different time scales contribute similarly and jointly to high flow events. While the most important drivers are similar across time scales, we find interesting variations for the second-most relevant drivers where evapotranspiration and surface soil moisture become more relevant towards longer time scales while deep soil moisture gets less relevant. Furthermore, for more extreme high flows we find a greater diversity of most important drivers across the considered catchments. Therefore, while moderate high flows are strongly associated with antecedent precipitation, the most extreme events can only be fully understood when considering a comprehensive selection of drivers. The spatial variations in the relevance of considered high flow drivers can be attributed to vegetation and terrain characteristics of the catchments. Our findings thereby illustrate that it is beneficial for flood monitoring and prediction to jointly consider several time scales and a comprehensive set of drivers physically related to streamflow dynamics. This way, identifying the relative importance of high flow generating mechanisms can reveal regional patterns of causes of floods in Europe and inform future model development. More recent model developments have focused on incorporating more processes into models. Our results based on multiple independent datasets provide an improved benchmark for evaluating all relevant hydrological processes in the model in a comprehensive manner. Further, given the relatively weak link between future precipitation and runoff changes, increasing attention has been paid to non-precipitation flood drivers (Brunner et al. 2021a). In this context, the framework introduced in this study provides a starting point to a data-driven investigation of possible future changes in high flow generation drivers and mechanisms globally to efficiently advance flood adaptation and resilience.



**Acknowledgements:**

M.L. and R.O. gratefully acknowledge funding from the German Research Foundation through Emmy Noether Grant No. 391059971. W.L. acknowledges a PhD scholarship from the China Scholarship Council as well as support from the International Max Planck Research School for Global Biogeochemical Cycles. S. O acknowledges the Brain Pool program funded by the Ministry of Science and ICT through the National Research Foundation of Korea (Grant NRF-2021H1D3A2A02040136).

This study is a contribution to the Euro-FRIEND project. We acknowledge streamflow data from the European water archive in cooperation with the EU-FP6 project WATCH and the E-OBS dataset established by the EU-FP6 project UERRA and the data providers in the ECA&D project (http://www.ecad.eu, accessed on 15 May 2020).

Furthermore, we thank Uwe Ehret, Kerstin Stahl and the Hydrology–Biosphere–Climate Interactions group in the Biogeochemical Integration Department of the Max Planck Institute for Biogeochemistry for fruitful discussions which helped to shape this study and the presentation of the findings. We are also grateful to Ulrich Weber for providing and preprocessing data.

**Competing interests**

The authors have no competing interests to declare.



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
