# Peer review of "Beyond precipitation: diversity of drivers of high river flows in European near-natural catchments"

_Hydrology and Earth System Sciences, 2022_

## Author Comment (AC1)

Reviewer 1:

This work presents an interesting investigation of floods across Europe. It aims to build on previous work by considering more factors as potential flood drivers (e.g. soil moisture at various depths, leaf area index, ET, and a metric precipitation variability.)

A1: We thank the reviewer for emphasizing the novelty of our study of investigating the mechanisms of high-flow generation.

The study concludes that:

1. This study provides a quantitative mapping of the importance of drivers of high river flow in near-natural European catchments.
2. […] that antecedent precipitation anomalies are the most important driver of high flows in most catchments.
3. [ …] In some other catchments snowmelt and soil moisture are found to be the most relevant drivers.
4. […] Moving beyond the state of the art we find a remarkable diversity of second-most important drivers across Europe. This includes vegetation-related drivers such as evapotranspiration.
5. […] Overall, observed daily high flow dynamics can be explained similarly well using drivers from the daily, weekly and monthly time scales. This indicates that mechanisms acting at different time scales contribute similarly and jointly to high flow events.
6. […] While the most important drivers are similar across time scales, we find interesting variations for the second-most relevant drivers where evapotranspiration and surface soil moisture become more relevant towards longer time scales while deep soil moisture gets less relevant. Furthermore, for more extreme high flows we find a greater diversity of most important drivers across the considered catchments.
7. […] Therefore, while moderate high flows are strongly associated with antecedent precipitation, the most extreme events can only be fully understood when considering a comprehensive selection of drivers.
8. […] The spatial variations in the relevance of considered high flow drivers can be attributed to vegetation and terrain characteristics of the catchments.
9. […] Our findings thereby illustrate that it is beneficial for flood monitoring and prediction to jointly consider several time scales and a comprehensive set of drivers physically related to streamflow dynamics.
10. […], Identifying the relative importance of high flow generating mechanisms can reveal regional patterns of causes of floods in Europe and inform future model development.
11. […] Moving beyond the state of the art we find a remarkable diversity of second-most important drivers across Europe. This includes vegetation-related drivers such as evapotranspiration.

All aspects 1-11, listed above are potentially relevant for publication in HESS. However, all these aspects also require some substantial consideration before I can recommend them for publication in HESS. My main concerns is:

The chosen method that relies on removing the seasonal cycle sounds potentially useful, but it is unclear to me how this should assign a dominant driver. In places where particular processes are underlying the flood response (e.g. snowmelt in NE Europe, this process is not considered important anymore in the analysis), and processes that physically can have no meaningful effect on floods at the given timescale (e.g. ET at daily timescale) are sometimes identified as most important process. The paper should manage to explain the attribution method (and the logic of removing the seasonal cycle) better to ensure the reader can trust these findings. The whole paper hinges on these findings, so it would be good if the reader can be better convinced of the presented approach.

A2: We thank the reviewer for pointing out his/her concern on our approach to remove mean seasonal cycles from the considered data streams. While our initial motivation for this was to investigate anomalies as society is probably adapted to (seasonal) mean conditions but the abnormal mechanisms are not exclusively addressed, we understand the reviewers comment that absolute quantities (e.g. water masses) might be more informative to study, and are more straightforward impact-relevant. Hence, we have updated our methodology and omitted the removal of mean seasonal cycles.

We update Figures 2-6 and section 2.2.3 accordingly. The results are actually not affected largely compared to results when removing mean seasonal cycles. Overall, there is a tendency for increased relevance of soil moisture at short time scales, and the attribution results related to vegetation types and terrain characteristics are clearer.

Some comments on the main conclusions

[1] Since the method seems to ignore relevant drivers and attribute irrelevant drivers, point 1 may not be not shown robustly

A3: We think that our updated methodology (see response A2) can address this point. If the reviewer would like to suggest additional relevant flood drivers to add to the extensive selection currently analyzed, we would be happy to implement.

[2-3] this seems in line with earlier work. Is there anything that we learn here that we did not know from previous studies? This may be useful to better highlight.

A4: The reviewer refers to a sentence in which we indicated an agreement with previous studies in the results and discussion section, while the subsequent sentences refer to new insights related to [4, 7, 8]. We agree that antecedent precipitation, snowmelt, and soil moisture are commonly illustrated as highly important flood drivers in previous studies. Obtaining similar results also helps to validate our methodology.

[4] If ET is really important (at daily timescales) this needs to be physically argued. Otherwise it is hard to be convinced by this finding.

A5: We agree with the reviewer that ET is expected to affect the water cycle and floods rather at longer time scales during which the daily fluxes can aggregate. This is already described in section 3.1. In order to avoid confusion we will adapt this sentence in the conclusions section to clarify that ET is relevant at weekly to monthly time scales, which is illustrated in Figure 4.

[5, 7] these are potentially very relevant and interesting findings, but they are only very briefly discussed and not very quantitatively shown in the paper. Can there be more explicit graphs/analyses that support these findings?

A6: We thank the reviewer for the positive feedback on our new findings.

We have implemented an additional analysis to support point [5]. We repeat the correlation analysis with all drivers from all three considered time scales to find the most relevant driver and time scale. This is done for each catchment. Then we aggregate these results across all catchments. We find that while preceding precipitation is still the main driver, the most relevant time scale varies between catchments, such that at the continental level all three considered time scales are found to be similarly relevant. This result will be included in our manuscript as an additional supplementary figure.

Moreover, when performing the multimodel inference analysis for each catchment considering the eight most important predictors across time scales found from the previous correlation analysis, we find that antecedent rainfall at daily or weekly is the main driver in different catchments. This highlights again the necessity of considering multiple time scales in high flow analyses.

For highlighting point [7], we provide more detailed information in section 3.1, as well as a new supplementary figure. Therein, we compare the mean fraction of high flow variance explained by only antecedent precipitation, as shown in the legend of Figure 3, with the mean explained fraction of variance of the full regression models considering all flood drivers. We find that the $R^2$ values are clearly higher for the full models, particularly for the more extreme floods, which quantitatively illustrates point [7].

[8] Ok, but what do we learn from this attribution? Can this be stated?

A7: We learn from this which are the main landscape characteristics and which modulate the relevance of flood drivers in space. This way, the main high flow drivers such as preceding precipitation and soil moisture affect high flows in different ways in different catchments. For example, the tree cover fraction and slope modulate the precipitation infiltration rate, and consequently the relevance of precipitation for flood events. We will clarify this point in section 3.2, as well as in the conclusions section.

[10] Can it be made a bit more explicit how models can benefit?

A8: Our main result is that a diversity of drivers and time scales needs to be considered to comprehensively understand, and accurately predict floods. This informs model development by suggesting alternative drivers and time scales to be more explicitly taken into account in

flood modelling in the future. And furthermore, our attribution findings stress the relevance of vegetation such that hydrological model development should ensure to appreciate and include the information of temporal and spatial vegetation dynamics. We will add these arguments to the conclusions section.

 [11] This diversity of drivers hinges on my main concern of the paper listed above.

A9: Please refer to response A2.

**Further comments**

- Considering ET and LAI as drivers of soil moisture on daily timescales seems nonsensical. How would these processes physically affect floods as ET and LAI will be tiny components of the total water balance during flood conditions on such timescales. Are their effects not already captured in considering soil moisture (which integrates the effects of E(T), as also is acknowledged in section 3.1)

A10: While we understand the reviewer's comment, we prefer to include the same set of drivers in the analyses for all time scales for consistency. As suggested by the reviewer, and described in response A5, ET is indeed not relevant at the daily time scale.

- It is unclear to me why the model selection leads to a set of near natural catchments, instead of just a set of catchments with simple to model behaviour (independent of the degree of human interference). I would be careful in qualifying these as near-natural.

A11: As we are using a versatile, conceptual model we assume that this scheme can reproduce streamflow whenever it is mainly controlled by meteorological variations instead of human interference. In addition, the streamflow dataset that we employ in our study describes the contained catchments as near-natural (Stahl et al. 2010). We will clarify this point in section 2.2.1.

- The choice of coarse spatial resolution of forcing data is understandable, but maybe problematic in the more mountains catchments. What are the potential consequences of this coarse data.

A12: The role of the spatial mismatch between the 0.25 degree flood driver data and the catchment-specific streamflow has been discussed in section 3.3 the limitations. The reviewer raises another valid point that this mismatch could be more problematic in mountainous regions. Figures 3 and 4 show that the spatial coherence of high flow drivers does not largely differ between mountainous areas like the Alps and their more flat surroundings. This suggests that the use of potential drivers with a 0.25˚ spatial resolution seems to be sufficient for our purpose. In addition, Figure 6 shows that basin area and terrain slopes play second-order roles in regulating the mechanisms of high flow generation. We will further clarify these points in section 3.3.

- Why are seasonal cycles removed, as these seasonal cycles might be important underlying drivers of the extreme events (i.e these are the ~sum of a seasonal cycle + an individual event on top of that). In places where processes are dominantly driven by a particular seasonal cycle (e.g. snowmelt in NE Europe and

large parts of Scandinavia, suddenly snow is not important anymore. How can you explain this to a reader?

A13: Motivated by the reviewer's comments, we have updated the methodology to keep the seasonal cycles. Please refer to response A2.

- Previous work across Europe also aggregates data across various time windows (e.g. Bloschl et al., 2017).

A14: This is true, but these studies do not compare the relevance of drivers considered at different time scales. To our knowledge, our study is the first analysis to do this systematically with a comprehensive set of drivers.

- When daily values are used, should rainfall on the date of the flood be chosen, or on the day before, or does this depend on the catchment size?

A15: This is an interesting point. After careful consideration we decided to keep our approach of considering the flood drivers on the day/week/month before the flood day. This enables us to understand and pinpoint flood drivers which are useful for predicting flood events, and we would not expect to have data on the flood day. We clarify this point in section 2.2.3.

- Figure 2: The font color of soil moisture layer 1 is hard to read.

A16: Adapted.

- Figure 3-4: this color classification is hard to read. It would also be useful to guide the reader in what the conclusion is of the Figure (within the caption).

A17: We will enlarge the colored points in the legend in Figures 3 and 4 to improve readability. Further, we will add a summary sentence of the findings in each figure to the caption.

- "Another interesting result is that the explained variance of high flows of the dominant drivers is similar across time scales. This indicates that studying drivers at different time scales is relevant to understand high flow dynamics, whereas daily, weekly and monthly time scales are similarly important. Multilayer soil moisture has a higher explained variance for events of the 99th percentile, suggesting the soil water storage is more relevant for the more extreme high flow generation." This is an interesting statement, but I think it requires some more analysis to conclude this. Right now this result is based on hand wavy interpretations of the results, and no formal quantitative comparison.

A18: Please refer to response A6 about new analyses that we provide to quantitatively study high flow drivers across different time scales. The second sentence in this statement will be revised to reflect the slight changes in the results in Figure 5 in relation to keeping the seasonal cycles.

References:

Stahl, K., Hisdal, H., Hannaford, J., Tallaksen, L. M., van Lanen, H. A. J., Sauquet, E., Demuth, S., Fendekova, M., and Jódar, J.: Streamflow trends in Europe: evidence from a dataset of near-natural catchments, Hydrol. Earth Syst. Sci., 14, 2367–2382.

---

## Author Comment (AC2)

Reviewer 2:

This paper presents a study about identifying different drivers of extreme flows over Europe by using observations from 250 near-natural catchments. It is well-written though;

B1: We thank the reviewer for appreciating our manuscript.

It basically uses some traditional and simple statistical approves say regression or correlation things to conduct some basic analysis between extreme flow and potential drivers or factors.

B2: While based on regressions, our multimodel inference methodology illustrated in Figure 2 is (i) powerful as it can reduce the risk of overfitting, and the effects of collinearities between predictor variables, and (ii) novel as it has not been used a lot in past high-flow analyses but has been successfully used in ecological studies such as Groemping et al., 2007, Fernández-Martíne et al., 2020, Jiao et al., 2021, considering the advantage mentioned in (i). We will clarify these points in the introduction.

Also, the paper did not transfer any new findings or new approach development. And no in-depth physical mechanisms have been introduced. Given that, it is not recommended for publication in this high-ranking journal.

B3: We disagree with the reviewer here. As stated in response B2 above, our statistical approach is a reliable tool to better understand the potential drivers of high flow events, which are so far often studied but no definitive conclusions. We do provide new insights aside from previous studies as stated in the conclusions section. Our study finds that (i) an interplay of multiple drivers induces extreme high flows, and (ii) drivers of high flow are relevant and need to be considered at different time scales. In addition, note that reviewer #1 recognizes the value of our study by mentioning e.g., "these are potentially very relevant and interesting findings" and "This work presents an interesting investigation of floods across Europe."

The reviewer gives a good point about physical mechanisms which we might write less clearly throughout the manuscript. We will list the hypotheses of each driver's physical process related to high-flow generations in the introduction, and expand the interpretation of physical mechanisms in the results and discussion section.

Concerns:

Line 15, it goes "Therefore it is imperative to understand their underlying physical mechanisms". After reading the paper, there is no information or analysis about the physical stuff.

B4: We note that physical linkages between high flows and the considered drivers have been discussed in several places in the manuscript, namely in sections 2.1.2, 2.2.4, 3.1, and 3.2.

In order to address the reviewer's concern we will introduce a table to synthesize and expand the physical mechanisms of potential high flow drivers as listed below.

Precipitation is a direct water input that contributes to soil moisture and runoff. But proportions of precipitation that can be partitioned into soil moisture and runoff depend on the soil moisture content as well as the precipitation intensity. For the precipitation intensity, we quantify such influence using the distribution of the rainfall over the considered time scale.

Snowmelt is another direct water input for runoff, and the input rate of snowmelt depends on the properties and amount of the snow, and meteorological conditions, e.g., temperature, radiation, and precipitation.

Soil moisture can contribute directly to runoff through the drainage and resulting baseflow, and additionally modulates the conversion of rainfall into streamflow.

Evapotranspiration (ET) affects soil moisture and thereby indirectly runoff. ET is controlled by meteorological conditions and also vegetation dynamics such as changes in leaf area index. Vegetation removes water from the soil through transpiration or interception evaporation on the vegetation surface.

Line 20. The paper goes like "And in these 11 articles the focus is mainly on regional and/or modelling studies, and they use some drivers for an explanation of the results rather than including them in the actual analysis.", which induces a justification of the current like "This leaves a knowledge gap in the joint understanding of a variety of observation-based controls of high river flows across continental-scale areas.". In fact, many existing studies have focused on identifying the possible contribution of extreme flow over the globe, including Europe as well. This paper should give a better explanation of why the current study should be done and why it is important.

B5: The current study is important because it jointly considers a comprehensive suite of potential high flow drivers (not only limited by hydrological processes but also combined with ecological processes) across different time scales to determine the main controls of high flows across Europe. Moreover, we for the first time analyse and attribute spatial changes in the relevance of vegetation and terrain characteristics on identified main controls. To do this, we take advantage of state-of-the-art datasets which benefit from recent advances in Earth observations and land surface modelling. We will clarify these points in the introduction section.

The section of 3.2 attribution analysis is quite loose and hard to explain. tree over fraction is the most important in explaining spatial patterns of the relevance of precipitation. This result is not new.

B6: Note that tree cover fraction does not only explain spatial variations in the relevance of rainfall for the occurrence of high flows, but also spatial variations in the relevance of snow melt and soil moisture. To our knowledge, we firstly illustrate the tree cover fraction as a main control out of other terrain or basin characteristics in influencing multiple drivers generating high flows.

While about the elevation and slope, the paper has no in-depth explanation about the potential relationship of the streamflow. And the basin area is of important to affect the effect of elevation and slope to flow. This also needs more physical explanations. Also see Fig. 6, the correlations of different time scales seem to be not consistent, even the direction (some positive or negative), this should be fully discussed.

B7: We agree with the reviewer that in previous results, except for the tree cover fraction, other attributes seem to have inconsistent patterns of regulating directions. For this, we have re-computed the attribution analysis of Figure 6 after keeping the seasonality in all our considered data streams (streamflow and potential drivers of high flows), to implement a suggestion from reviewer #1 by considering the potential seasonal influence of high flow drivers (see response A2). Interestingly, the updated figure confirms the previous results of tree cover fraction as the main control, and shows more consistent patterns for other second-order drivers across time scales such as basin areas. The new figure shows that rainfall is more relevant for high flows in small catchments where the entire area can be affected by extreme rainfall at the same time. Soil moisture is more relevant for larger catchments as these are typically not simultaneously affected by strong rainfall. Further, soil moisture is more relevant in low-elevation catchments where soils can easily be water logging, while rainfall and snowmelt

are more relevant in high-elevation catchments. Finally, soil moisture is more relevant in catchments with strong slopes, as this favors lateral flow, while the opposite is observed for precipitation.

We will update Figure 6, and include these arguments in section 3.2.

This paper in fact did some basic statistical analysis, while all paper gives an impression about trying to link the physical mechanisms to the changes of flows. Yet, this is no physical analysis across the paper and no physical explanations but some statistical analysis.

B8: As recognized by the reviewer, we are linking physical mechanisms to drivers of high flows in order to explain their relevance. We do this through a data-driven approach which detects emergent relationships from observation-based data and interpret in a process-oriented way. Therefore, our approach nicely complements model-based analyses which can more explicitly study physical processes but are limited by a potentially inaccurate and/or incomplete representation of processes in the model.

We will clarify this point in the conclusions section.

References:

Groemping, U. Relative importance for linear regression in R: the package relaimpo. J. Stat. Softw. 17, 1–27 (2007).

Fernández-Martínez, M. et al. The role of climate, foliar stoichiometry and plant diversity on ecosystem carbon balance. Glob. Change Biol. 26, 7067–7078 (2020).

Jiao, W., Wang, L., Smith, W. K., Chang, Q., Wang, H., & D'Odorico, P. (2021). Observed increasing water constraint on vegetation growth over the last three decades. Nature Communications, 12(1), 3777.